# Single and Repeated Exposure to Cannabidiol Differently Modulate BDNF Expression and Signaling in the Cortico-Striatal Brain Network

**DOI:** 10.3390/biomedicines10081853

**Published:** 2022-08-01

**Authors:** Francesca Mottarlini, Marco Fumagalli, Fernando Castillo-Díaz, Stefano Piazza, Giorgia Targa, Enrico Sangiovanni, Barbara Pacchetti, Mikael H. Sodergren, Mario Dell’Agli, Fabio Fumagalli, Lucia Caffino

**Affiliations:** 1Department of Pharmacological and Biomolecular Sciences, Università degli Studi di Milano, Via Balzaretti 9, 20133 Milano, Italy; francesca.mottarlini@unimi.it (F.M.); marco.fumagalli3@unimi.it (M.F.); fer.castillo@gmail.com (F.C.-D.); stefano.piazza@unimi.it (S.P.); giorgia.targa@unimi.it (G.T.); enrico.sangiovanni@unimi.it (E.S.); mario.dellagli@unimi.it (M.D.); lucia.caffino@unimi.it (L.C.); 2Curaleaf International, London EC2A 2EW, UK; bp@curaleafint.com (B.P.); m.sodergren@imperial.ac.uk (M.H.S.); 3Medical Cannabis Research Group, Department of Surgery and Cancer, Imperial College London, London SW7 2AZ, UK

**Keywords:** cannabidiol, prefrontal cortex, striatum, BDNF

## Abstract

Cannabidiol (CBD) is a phytocannabinoid contained in the Cannabis sativa plant, devoid of psychotomimetic effects but with a broad-spectrum pharmacological activity. Because of its pharmacological profile and its ability to counteract the psychoactive Δ^9^-tetrahydrocannabinol (Δ9THC), CBD may be a potential treatment for several psychiatric and neurodegenerative disorders. In this study, we performed a dose−response evaluation of CBD modulatory effects on BDNF, a neurotrophin subserving pleiotropic effects on the brain, focusing on the cortico-striatal pathway for its unique role in the brain trafficking of BDNF. Male adult rats were exposed to single and repeated CBD treatments at different dosing regimen (5, 15, and 30 mg/kg), to investigate the rapid modulation of the neurotrophin (1 h after the single treatment) as well as a potential drug-free time point (24 h after the repeated treatment). We show here, for the first time, that CBD can be found in the rat brain and, specifically, in the medial prefrontal cortex (mPFC) following single or repeated exposure. In fact, we found that CBD is present in the mPFC of rats treated either acutely or repeatedly with the phytocannabinoid, with a clear dose−response profile. From a molecular standpoint, we found that single, but not repeated, CBD exposure upregulates BDNF in the mPFC, while the repeated exposure increased BDNF only in the striatum, with a slight decrease in the mPFC. Together, these data reveal a CBD dose-dependent and anatomically specific modulation of BDNF, which may be functionally relevant and may represent an added value for CBD as a supplement.

## 1. Introduction

Cannabidiol (CBD) is one of the most abundant phytocannabinoids present in the Cannabis sativa plant, devoid of psychotomimetic effects [1]. In addition, CBD antagonizes several of the psychoactive effects of the Δ9-tetrahydrocannabinol (Δ9-THC), the major psychoactive compound of Cannabis sativa [2,3]. CBD exhibits a broad-spectrum pharmacological profile that makes it a potential treatment for several psychiatric and neurodegenerative disorders [4,5]. In fact, while CBD displays low affinity for cannabinoid CB1 and CB2 receptors [6], it can act as a negative allosteric modulator at these receptors [7,8]. In addition, its mechanism of action involves, at least, inhibition of anandamide hydrolysis [9], as well as action at the vanilloid receptor 1 (TRPV1) [9], the serotoninergic 5-HT1A receptor [10], and PPARγ receptors [11]. Friedman et al., reviewed the pharmacology of cannabinoids in neurological disorders, underlining the criticisms related to the bioavailability of these highly lipophilic molecules, such as the way of administration. With regards to CBD, the authors concluded that the bioavailability of oral administration in humans is poor (6–19%) and variable, according to several clinical studies [12,13]. For this reason, many in vivo studies concerning the effects of CBD in the central nervous system (CNS), including ours, considered other ways of administration, such as intraperitoneal (i.p.) injection. Regardless of the mode of administration, a clear characterization of the brain distribution of CBD and of its neuroplastic effects is still lacking.

Brain-derived neurotrophic factor (BDNF) belongs to a group of proteins termed neurotrophins, which play a pleiotropic role in the CNS. Besides its role during neurodevelopment and its neuroprotective properties, BDNF and its high affinity receptor TrkB are crucial regulators of neuroplasticity [14]. BDNF-induced activation of downstream signaling cascades regulates different cellular processes including neuroprotection, cognition, and stress coping [15]. Furthermore, modulation of BDNF levels may be critical for treatment of both psychiatric [16] and neurodegenerative disorders [17].

The organization of the BDNF gene is complex, exhibiting several 5′ non-coding exons, each with a separate promoter region that triggers the transcription of a common 3′ exon and encoding for the same protein [18]. The functions of these multiple variants have not yet been fully clarified but, indeed, specific transcripts may undergo different intracellular targeting. Thus, it appears that BDNF function is controlled in a dynamic manner not only at the transcriptional and translational level, but also via specific mRNA targeting as well as processing and secretion of its protein [19,20]. In fact, the cleavage of the BDNF precursor gives rise to the mature form of the neurotrophin (mBDNF, 14 kDa), which is anterogradely transported to its target neurons [21,22]. This is an important point in the overall regulation of the BDNF system as, for instance, BDNF is not synthesized in the striatum. After its release, BDNF binds to its high-affinity receptor TrkB that, after autophosphorylation, stimulates downstream pathways, primarily phosphoinositide 3-kinase (PI3-K) and mitogen-activated protein kinase (MAPK) pathways. Several botanicals and natural compounds, including CBD, have been shown to modulate neuroplasticity acting on BDNF levels [23] in animal models of several disorders such as depression [24], schizophrenia [25], addiction [26], ischemia [27], and Alzheimer’s disease [28]. However, information on the fine-tuned regulation of BDNF in the action of CBD is still fragmentary.

Accordingly, we performed a dose−response study in rats to evaluate the rapid modulation (1 h) of the neurotrophin as well as its downstream signaling following a single injection of CBD. In addition, we exposed rats to repeated CBD treatment (five consecutive injections, once a day) with sacrifice 24 h after the last injection, to investigate a potential drug-free time point, focusing our attention on CBD-induced modulation of BDNF in the cortico-striatal pathway.

## 2. Materials and Methods

### 2.1. Animals

Adult male Sprague Dawley rats (Charles River, Calco, Italy) weighing 280 to 300 g on arrival were housed in groups of 2 in standard polycarbonate cages under standard laboratory conditions of temperature (21 ± 1 °C), humidity (50–60%), and artificial light (from 07:00 to 19:00 h). [29]. All animal procedures were conducted at the Department of Pharmacological and Biomolecular Sciences at the University of Milan, and carried out in accordance with the principles set out in the following laws, regulations, and policies governing the care and use of laboratory animals: Italian Governing Law (D.lgs 26/2014; Authorization n.19/2008-A issued 6 March 2008, by Ministry of Health); the NIH Guide for the Care and Use of Laboratory Animals (2011 edition) [30]; and EU directives and guidelines (EEC Council Directive 2010/63/UE). All efforts were made to minimize animal suffering and to keep the lowest number of animals used. The experiments have been reported in compliance with the ARRIVE guidelines.

### 2.2. Drug Preparation

CBD extracted from Cannabis plants was provided by Curaleaf International, UK, as a purified powder. Purity for CBD was above 95%; HPLC traces are provided as CBD powder was dissolved in a vehicle of Tween-80 (2%) and saline (NaCl 0.9%), protected from light, in agitation at 60 °C (avoiding boiling) until the complete mixing. The dissolved CBD was freshly prepared immediately prior to injection, and it was administered at different concentrations: 5 mg/kg, 15 mg/kg, and 30 mg/kg.

### 2.3. Experiments

Upon arrival, all the animals were habituated in the facility before starting any procedure for one week. Rats were left undisturbed for two days and then they were handled for 5 days before the injections; such manipulation was performed to avoid any potential bias due to stress-related effects. After this period of acclimation to the facility and the procedures, a total of 48 animals were divided into two separate sets of rats for acute (Experiment 1) and repeated (Experiment 2) treatments with CBD. In Experiment 1, a subset of 32 male Sprague Dawley rats was exposed to acute treatment of different CBD doses and was further subdivided into four experimental groups: (1) control group receiving a single i.p. injection of saline (*n* = 8), (2) single i.p. injection of CBD 5 mg/kg (*n* = 8), (3) single i.p. injection of CBD 15 mg/kg (*n* = 8), and (4) single i.p. injection of CBD 30 mg/kg (*n* = 8). The animals were sacrificed one hour after the acute treatment with saline or CBD. In Experiment 2, a subset of 32 male Sprague Dawley rats was subdivided into four experimental groups: rats exposed to (1) repeated saline (*n* = 8), or CBD treatments at the dose of (2) 5 mg/kg (*n* = 8), (3) 15 mg/kg (*n* = 8), or (4) 30 mg/kg (*n* = 8) for seven consecutive days. Animals were sacrificed twenty-four hours after the last treatment with saline or CBD. After decapitation, brains from Experiment 1 and 2 were rapidly removed, and the medial prefrontal cortices (mPFC, defined as Cg1, PL, and IL subregions, corresponding to plates 5–9) and striatum tissues (caudate putamen, corresponding to plates 10–25) were immediately dissected from 2 mm thick slices following the coordinates of the Rat Brain Atlas of Paxinos and Watson [31], then frozen on dry ice and stored at −80 °C for subsequent molecular analysis.

### 2.4. Plasma Preparation for CBD Measurements

Trunk blood from each rat was promptly collected after decapitation in tubes containing sodium citrate 3.8% (200 μL × 2 mL of blood collected) as anticoagulant agent. Plasma was separated by centrifugation (6500× *g* for 20 min) and stored at −20 °C for future molecular analysis. Plasma samples were prepared for CBD quantification as follows: 25 μL of the internal standard [cannabigerol (CBG) 3 ng/μL] was added to 150 μL of each plasma sample. Protein precipitation was performed by adding 100 μL of acetonitrile, while the extraction of CBD was performed with 1 mL of hexane. The samples were mixed by vortex and centrifuged at 10,000 rpm for 10 min, 900 μL of the supernatant was transferred into a new 1.5 mL tube, and the solvent was removed under nitrogen gas flow. The residue was reconstituted in 150 μL of methanol. Each sample was mixed and centrifuged at 10,000 rpm for 5 min. The final supernatant was used for the LC–MS/MS analysis.

### 2.5. Brain Tissue Preparation for CBD Measurements

The extraction of CBD from the mPFC was performed by using the whole protein homogenate extracted from each sample (see Protein Extracts preparation below). Ten μL of the internal standard (CBG 3 ng/μL) was added to 250 μL of each protein homogenate. The samples were mixed by vortex and centrifuged at 10,000 rpm for 10 min, 900 μL of the supernatant was transferred into a new 1.5 mL tube, and the solvent was removed under nitrogen gas flow. The residue was reconstituted in 150 μL of methanol. Each sample was mixed and centrifuged at 10,000 rpm for 5 min. The final supernatant was used for the LC–MS/MS analysis [32].

### 2.6. LC–MS/MS Analysis

HPLC was performed through an Exion LCTM AC System (AB Sciex, Foster City, CA, USA) composed of a vacuum degasser, a double plunger pump, a cooled autosampler, and a temperature-controlled column oven. The MS/MS analysis was carried out with a Triple QuadTM 3500 system (AB Sciex, Foster City, CA, USA). The analytes were separated on a Synergi 4 um Hydro-RP 80 A LC Colum 150 × 4.6 mm (Phenomenex, Torrance, CA, USA) with a mobile phase composed of 0.1% formic acid in water (A) and methanol (B) at a rate flow of 0.800 mL/min. The chromatographic gradient is described below in Table 1.

The injection volume was 10 μL for each sample. Mass spectrometric detection was done in negative ionization (ESI) mode, and the parameters were set as follows: curtain gas at 30 psi, ionization voltage at −4500 V, source temperature at 500 °C, and nebulization gas 1 and nebulization gas 2 at 50 psi. The optimized compound-dependent MS/MS parameters (declustering potential, entrance potential, collision energy, and collision cell exit potential) were obtained, in multiple-reaction-monitoring (MRM) mode, by a separate infusion of the analyte (CBD) and the internal standard (CBG). The analyte and the internal standard were analyzed by using the following mass transitions: 313/245 (CBD), 315/136 (CBG). The LC–MS/MS system was controlled by AB Sciex Analyst (version 1.7) software.

The Limit of Detection (LOD) of CBD was detected by multiple injections in LC-MS of serial dilutions of the sample. LOD was found 1.2 pg. This can be considered the lowest concentration of CBD that the instrument is able to detect.

### 2.7. RNA Preparation and Real-Time Polymerase Chain Reaction

Total RNA from mPFC was isolated by single-step guanidinium isothiocyanate/phenol extraction using PureZol RNA isolation reagent (Bio-Rad Laboratories, Segrate, Milan, Italy) according to the manufacturer’s instructions and quantified by spectrophotometric analysis. Following total RNA extraction, the samples were processed for real-time reverse transcription polymerase chain reaction (real time RT-PCR) to assess mRNA levels. Briefly, an aliquot of each sample was treated with DNase to avoid DNA contamination. RNA was analyzed by TaqMan qRT-PCR instrument (CFX384 real time system, Bio-Rad Laboratories) using the iScriptTM one-step RT-PCR kit for probes (Bio-Rad Laboratories). Samples were run in 384-well formats in triplicate as multiplexed reactions. Data were analyzed with the comparative threshold cycle (∆∆Ct) method using *36B4* as reference gene [33]. Primers and probe for *Bdnf exon IV* and *VI* were purchased from Applied Biosystems (*Bdnf exon IV*: ID Rn01484927_m1 and *Bdnf exon VI*: ID Rn01484928_m1). Primers and probe for total *Bdnf* and *36B4* were purchased from Eurofins MWG-Operon. Their sequences are shown below:*total Bdnf*: forward primer 5′-AAGTCTGCATTACATTCCTCGA-3′, reverse primer 5′-GTTTTCTGAAAGAGGGACAGTTTAT-3′, probe 5′- TGTGGTTTGTTGCCGTTGCCAAG-3′;*36B4*: forward primer 5′-TTCCCACTGGCTGAAAAGGT-3′, reverse primer 5′-CGCAGCCGCAAATGC-3′, probe 5′-AAGGCCTTCCTGGCC GATCCATC-3′.

### 2.8. Protein Extracts Preparation and Western Blot Analyses

Proteins from mPFC and striatum were homogenized in a glass−glass potter in cold 0.32 M sucrose buffer pH 7.4 containing 1 mM HEPES, 0.1 mM PMSF, in presence of commercial cocktails of protease and phosphatase (Sigma-Aldrich, Milan, Italy) inhibitors and then sonicated. Total proteins were measured in the whole homogenate and quantified according to the Bradford Protein Assay procedure (Bio-Rad, Milan, Italy), using bovine serum albumin as calibration standard, and stored at −20 °C for subsequent molecular analysis. Western blots (WB) were run using sodium dodecyl sulfate −8% polyacrylamide gel under reducing conditions as previously described [34] on the whole homogenate lysate (10 μg) of mPFC and striatum and then electrophoretically transferred (dry transfer) onto nitrocellulose membranes (GE Healthcare). The strips of nitrocellulose membrane close to the molecular weights at which the bands of the protein of interest were expected were cut from the entire squared blot (full areas) as suggested by their specific molecular weight and the information present in the datasheet of the antibody. Blots were blocked for 1 h at room temperature (25 ± 2 °C) with I-Block solution (Life Technologies, Monza, Italy) in TBS + 0.1% Tween-20 buffer and washed with TBS + 0.1% Tween-20 buffer. The conditions of the primary antibodies were the following: anti mBDNF (1:1000, Icosagen, Tartu, Estonia, cod: 327-100), anti phospho-TrkB Tyr706 (1:500, Novus Biologicals, Littleton, CO, USA, cod: NBP2-54764), anti phospho-Akt Ser473 (1:1000, Cell Signaling Technology, Danvers, MA, USA, cod: 9271), anti phospho-ERK2 Thr185/Tyr187 (1:1000, Cell Signaling Technology, cod: 4370), anti-total TrkB (1:500, Cell Signaling Technology, cod: 4603), Akt (1:1000, Cell Signaling Technology, cod: 9272), ERK2 (1:5000, Cell Signaling Technology, cod: 4695), and anti β-actin (1:10.000, Sigma-Aldrich, cod: A5441). Results were standardized to β-actin control protein, which was detected by evaluating the band density at 43 kDa. Immunocomplexes were visualized by chemiluminescence using the Chemidoc MP Imaging System (Bio-Rad Laboratories) after 2–3 min of enhanced chemiluminescence substrate (ECL) exposure (Cyanagen Srl, Bologna, Italy). Activation of the proteins investigated were expressed as a ratio between the phosphorylated and the respective total forms and analyzed. Gels were run two times each, and the results represent the average from two different runs. We used a correction factor to average the different gels: correction factor gel B = average of (OD protein of interest/OD β-actin for each sample loaded in gel A)/(OD protein of interest/OD β-actin for the same sample loaded in gel B) [35].

### 2.9. Statistical Analysis

Data were collected in individual animals and are presented in bar graphs as means of 8 independent determinations ± standard errors (SEM). Shapiro−Wilk and the Kolmogorov-Smirnov tests were employed to determine normality of residuals (see Appendix A). Molecular results from the single CBD treatment as well as CBD content measurements in plasma and mPFC and mRNA and protein level determinations, normally distributed, were analyzed by ordinary one-way ANOVA followed by Tukey’s post-hoc test, or for repeated CBD treatments by unpaired Student’s *t*-test. Data with a non-normal distribution were analyzed by the Kruskal−Wallis one-way ANOVA for ranks followed by Dunn’s multiple comparisons test, or by the Wilcoxon−Mann−Whitney test. Outlier calculation was performed with Grubb’s test on the free platform GraphPad by Dotmatics. Statistical significance was assumed at *p* < 0.05.

## 3. Results

Figure 1 shows CBD plasma levels following single (Figure 1a) or repeated injections (Figure 1b) with 5, 15 or 30 mg/kg of CBD. The concentration of CBD in the plasma increased dose-dependently under both experimental conditions (Figure 1a: KW = 18.24, *p* = 0.0001; CBD 5 mg/kg = +32.53 ng/mL; CBD 15 mg/kg = +79.78 ng/mL vs. 5 mg/kg, *p* = 0.0327; CBD 30 mg/kg = +181.998 ng/mL vs. 5 mg/kg, *p* < 0.0001; Figure 1b: F(2,21) = 102.4, *p* < 0.0001; CBD 5 mg/kg = +2.04 ng/mL; CBD 15 mg/kg = +10.51 ng/mL vs. 5 mg/kg *p* = 0.0010, CBD 30 mg/kg + 23,96 ng/mL vs. 15 mg/kg, *p* < 0.0001).

Figure 2 shows CBD cortical levels following single (Figure 2a) or repeated injections (Figure 2b) with 5, 15, or 30 mg/kg of CBD. At variance from plasma CBD levels, it appears that cortical CBD concentration rises significantly over untreated rats only at the highest dose employed (i.e., 30 mg/kg) under both experimental conditions (Figure 2a: KW = 12.89, *p* = 0.0016; CBD 30 mg/kg = +0.31 ng CBD/ng protein vs. 5 mg/kg, *p* = 0.0010; Figure 2b: KW = 17.90, *p* = 0.0001; CBD 30 mg/kg + 0.10 ng CBD/ng protein vs. 5 mg/kg, *p* = 0.0001; CBD 30 mg/kg + 0.09 ng CBD/ng protein vs. 15 mg/kg, *p* = 0.0070). Of note, CBD was undetectable at the 5 mg/kg dose following both single and repeated administrations.

We next evaluated the dose-dependent effects of acute CBD exposure on the gene expression levels of *total Bdnf* and related exons in the mPFC. In particular, we analyzed the main *Bdnf* exons, i.e., *exon IV*, the most abundant exon of somatic origin, and *exon VI*, which is expressed at dendrite level. Figure 3 illustrates the effects of a single injection of 5, 15, and 30 mg/kg of CBD on the gene expression levels of *total Bdnf* and its exons. While *total Bdnf* as well as *exon VI* gene expression levels are unchanged (Figure 3a: F(3,27) = 1.553, *p* = 0.224; Figure 3c: F(3,27) = 1.980, *p* = 0.1407), the transcription of *exon IV* is significantly upregulated in the mPFC, but only at the highest dose employed (Figure 3b: KW= 10.24, *p* = 0.0166, + 21.14% vs. saline *p* = 0.0173).

We then analyzed protein levels of the mature form of the neurotrophin BDNF (mBDNF). As shown in Figure 4, acute CBD treatment up-regulates mBDNF levels in the whole homogenate of the mPFC at 30 mg/kg (Figure 4a: F(3,27) = 3.361, *p* = 0.0333, + 30.00% vs. saline *p* = 0.0230), but not at the lower doses used. Similarly, we found increased phosphorylation of the BDNF high-affinity receptor TrkB in Tyr(Y)706 (Figure 4b: F(3,28) = 7.893, *p* = 0.0006, + 43.63% vs. saline *p* = 0.0035), with no changes in its total expression (Figure 4c: KW 4.440, *p* = 0.2177). Of note, the levels of TrkB receptor activation, expressed as the ratio between the phosphorylated and non-phosphorylated (pTrkB/TrkB), are increased only following the single injection of CBD 30 mg/kg (Figure 4d: KW= 17.82, *p* = 0.0005, + 15.25% vs. saline *p* = 0.0220), with no effect at the lower doses.

Then, to investigate whether alterations in the BDNF-TrkB system induced by a single CBD exposure would engage the recruitment of BDNF downstream pathways, we analyzed the expression and phosphorylation of Akt and ERK2 effectors (Figure 5). As shown in Figure 5, while no significant changes are observed either in the phosphorylated (Ser473) (Figure 5a: F(3.28) = 2.907, *p* = 0.0521) or in the total form of Akt (Figure 5b: KW = 6.804, *p* = 0.0784), their ratio expressed as pAkt/Akt is significantly increased following the acute CBD treatment at 30 mg/kg (Figure 5c: F(3.28) = 2.959, *p* = 0.0494, + 33.38% vs. saline *p* = 0.0328). Conversely, the analysis of ERK2 phosphorylation (Thr185/Tyr187) does not show any relevant changes (Figure 5d: KW= 8.619, *p* = 0.0348; Figure 5f: KW= 1.054, *p* = 0.7882), whereas, despite that the ANOVA analysis of ERK2 expression is statistically significant, the multiple comparisons test does not show any relevant change (Figure 5e: KW = 4.662, *p* = 0.1983).

Based on the results observed following the single exposure of CBD, showing a significant effect on BDNF and its downstream signaling only at the highest dose employed, we decided to further analyze the neuroplastic effect of a repeated treatment with the phytocannabinoid employing only the highest dose. We found that repeated exposure to CBD (30 mg/kg) did not cause any change in the gene expression levels of *total Bdnf*, and *exon IV* and *VI* in the mPFC (Figure 6a: −10.75% vs. saline, t(14) = 1.224, *p* = 0.2410; Figure 6b: −9% vs. saline, t(14) = 0.8836, *p* = 0.3918; Figure 6c: −9.75% vs. saline, t(14) = 1.027, *p* = 0.3219). Of note, despite that no alterations are present in the mRNA levels of Bdnf and related exons, repeated CBD exposure reduces slightly, but significantly, mBDNF protein levels (Figure 7a: −11.75% vs. saline, t(14) = 2.443, *p* = 0.0284) and pTrkB/TrkB ratio (Figure 7d: −14.25% vs. saline, t(14) = 2.708, *p* = 0.0170).

Interestingly, we found that the pAkt/Akt ratio is reduced (Figure 8c: −33.13% vs. saline, t(14) = 2.959, *p* = 0.0104), in line with reduced levels of Akt phosphorylation in Ser473 (Figure 8a), whereas no changes are observed in total Akt levels (Figure 8b: +9% vs. saline, t(14) = 0.6370, *p* = 0.5344). As previously shown after a single injection, the analysis of ERK2 does not show any alteration following repeated CBD exposure when compared to saline-treated animals (Figure 8d: −0.88% vs. saline, t(14) = 0.0594, *p* = 0.9535; Figure 8e: −7% vs. saline, U = 17, *p* = 0.1304; Figure 8f: +7% vs. saline, t(14) = 0.5305, *p* = 0.6041).

BDNF protein is known to undergo anterograde transport from the mPFC toward striatum [21]. In line with the reduction observed in the mPFC, we analyzed the BDNF-TrkB system in the striatum. Of note, accordingly, mBDNF protein levels are increased following repeated exposure to 30 mg/kg of CBD (Figure 9a: +36.5% vs. saline, U= 0, *p* = 0.0002). No changes are observed in the TrkB receptor levels (Figure 9b–d), either in the phosphorylated (Figure 9b: −12.75% vs. saline, U= 20, *p* = 0.3969) or in the total form of TrkB receptor (Figure 9c: −8.37% vs. saline, t(14) = 1.454, *p* = 0.1680) as well as in the pTrkB/TrkB ratio (Figure 9d: −7.5% vs. saline, U = 19, *p* = 0.3357). The evaluation of BDNF downstream effectors Akt and ERK2 revealed a significant increase in pAkt (Ser473) (+29.13% vs. saline, t(14) = 4.599, *p* = 0.0004) and total Akt (+24.88% vs. saline, t(14) = 5.550, *p* < 0.0001) as shown in Figure 10a,b, respectively, with no changes in the pAkt/Akt ratio (Figure 10c: (+3.75% vs. saline, t(14) = 0.7222, *p* = 0.4821). In line with our previous observations, no changes are detected for ERK2 (Figure 10d: −12.38% vs. saline, t(14) = 1.579, *p* = 0.1366; Figure 10e: +1.38% vs. saline, t(14) = 0.1393, *p* = 0.8912; Figure 10f: −8.13% vs. saline, U= 20, *p* = 0.2345).

## 4. Discussion

We show here, for the first time, that following single or repeated exposure, CBD can be found in the rat brain and, specifically, in the medial prefrontal cortex (mPFC). In fact, we found that CBD is present in the mPFC of rats treated either acutely or repeatedly with the phytocannabinoid, with a clear dose−response profile. In fact, whereas CBD cannot be detected at the lowest dose of CBD, i.e., 5 mg/kg, it is measurable at the dose of 15 mg/kg, reaching its peak at the maximal dose tested (30 mg/kg). Similarly, with the brain, a clear dose−response effect of CBD exposure was observed when measuring its levels in the rat plasma. In fact, it is barely detectable at 5 mg/kg of CBD, whereas its concentration increases dose-dependently at the other doses employed, reaching concentrations that are indeed much higher when compared with the brain levels. Again, we observed a significant reduction in CBD following repeated exposure when comparing plasma concentrations with those of single exposure. Taken together, these results indicate that CBD is already bioavailable within 1 h following the single injection and it is still detectable 24 h after the repeated administration, albeit at much lower concentration. Other authors investigated the impact of different methods (pulmonary, oral, and subcutaneous) and dosage of acute administration, on brain and serum level of CBD in rats. For this reason, the comparison among previous works and ours with respect to brain uptake is hard to perform. Independently of the route of administration, doses comparable to those used in our study (10 mg/kg) led to relevant brain uptake. Hlozek et al., showed a peak of concentration of 200 ng/g at 2 h after oral administration that was enhanced by the feeding state of the animals [36]. Another study observed a high brain uptake after acute subcutaneous injection of 10 mg/kg of CBD in mice with a peak reached after 1 h [37]. Again, Deiana et al., evaluated the pharmacokinetics of a high acute dose of CBD (120 mg/Kg) in rats (i. p.), thus measuring a Tmax of 60–120 min and relevant amount of CBD still after 24 h at brain level [38]. To the best of our knowledge, no previous studies regarding repeated exposure to CBD were conducted.

In line with the presence of the phytocannabinoid in the mPFC following a single exposure, we found that a single dose (30 mg/kg) of CBD is sufficient to upregulate Bdnf exon IV, the most abundant variant of the BDNF gene, which is paralleled by a similar increase in cortical mBDNF and TrkB. Such up-regulation drives the selective activation of the PI3K pathway (i.e., Akt). These effects might be beneficial for the overall regulation of cell homeostasis, potentially fostering the ability to promote synaptic transmission and plasticity, neuroprotection, and activity-dependent structural remodeling [14].

Conversely, a slight decrease was observed in the levels of mBDNF paralleled by reduction in TrkB and Akt activation following repeated exposure to CBD in the mPFC. Such a variable profile is not surprising when examining the neurotrophin expression [39,40]. To further strengthen this concept, we have also shown that an opposite modulation of BDNF can dissect the antidepressant from the reinforcing properties of ketamine [41]. These lines of evidence suggest that a more prolonged treatment with CBD should be performed to cause BDNF up-regulation.

As already revised by Lucas et al. [42] and Ujvàry et al. [43], CBD is mainly metabolized by iso-enzymes CYP2C19 and CYP3A4 at hepatic level. Several studies involving rodent models showed that CBD is mainly excreted in the intact or glucuronide form, while the major metabolites are hydroxylated derivatives and their glucuronide conjugates. For this reason, hepatic passage could be involved in the reduction in CBD plasmatic concentration during repeated exposure. However, the pharmacology of hydroxylated metabolites is still poorly investigated.

Overall, these data indicate that a single dose of 30 mg/kg is sufficient to trigger the activation of BDNF and its downstream signaling in the mPFC, whereas repeated exposure to the same dosing regimen downregulates the neurotrophin system. These data highlight the tight dependence of BDNF modulation upon CBD levels in the mPFC. In fact, it appears that when CBD is available in the mPFC in appreciable concentrations, the BDNF system is activated, such as after the single treatment. This finding may also suggest that, at least at the 30 mg/kg dose, which is indeed not the highest used in the literature, CBD may mediate, through BDNF, some short-term benefits for the brain by facilitating synaptic transmission, rather than long-term benefits. One potential outcome of the increase observed following a single exposure to BDNF relies on the possibility that it may contribute toward setting up a proper adaptive response of neural cells in response to environmental challenges, be they positive or adverse. For instance, we have demonstrated that exposure to stress prevents the ability to mount a neuroadaptive response to adverse external stimuli via up-regulation of cortical BDNF expression [44]. However, it is interesting to note that a single stress also favored the performance in a single cognitive test through transient up-regulation of cortical Bdnf mRNA levels [45]. Based on these lines of evidence, there is the possibility that CBD-induced up-regulation of cortical BDNF expression may favor cell coping under similar situations.

Another interesting finding of our experiments derives from the evidence that BDNF expression is elevated in the rat striatum following repeated exposure. It is well established that the striatum lacks Bdnf messenger RNA and that the neurotrophin is supplied to the striatum through anterograde transport from the mPFC [21]. Notably, we found a reduction in mBDNF in the mPFC and a significant increase in the striatum, potentially supporting an increased anterograde trafficking of BDNF, mediated by CBD. Interestingly, it appears that CBD has also activated the downstream BDNF pathway mediated by Akt, thus mediating the intracellular signaling cascade promoted by the neurotrophin. Notably, it has been previously shown that CBD leads to stronger connectivity between prefrontal cortex and striatum in humans, an effect that may perhaps be due to BDNF trafficking [46]. Evidence also exists that BDNF is critical for the survival of striatal neurons in animal models of Huntington’s disease [47]; therefore, repeated treatment with CBD, through BDNF up-regulation, may represent a potential strategy to rescue, at least partially, striatal neurons from degeneration. BDNF is also important for the survival of striatal GABA neurons [48]. This is crucial in view of the notion that dysfunction in cortical and subcortical GABAergic pathways characterize, among others, the pathophysiology of schizophrenia [49]. It is in fact established that, in the striatum, GABA is pivotal for the regulation of overactivity of excitatory neurotransmissions as well as memory functions, which are perturbed in schizophrenic patients [50,51]. Taken together, these data suggest that CBD-induced increase in striatal BDNF may be functionally relevant and may represent an added value for CBD as supplement.

## 5. Conclusions

In conclusion, we have demonstrated that CBD can be detected in the plasma and mPFC, following single or repeated injections. In both districts, CBD is detected following a specific dose−response profile. Further, we showed that CBD can modulate BDNF expression in a manner that depends upon the length of the treatment and following a specific anatomical pattern. Accordingly, our data are likely to reflect the targeting of specific neuroplastic processes in the cortico-striatal pathway rather than an interference with specific neurotrophic responses.

## Figures and Tables

**Figure 1 biomedicines-10-01853-f001:**
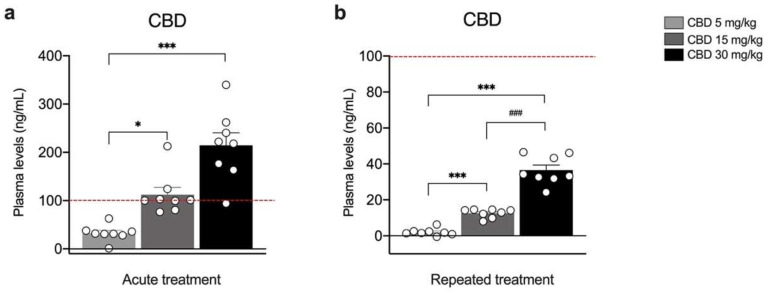
Concentration of CBD in the plasma of rats treated with a single (**a**) or repeated (**b**) CBD administration at the dose of 5 mg/kg, 15 mg/kg, or 30 mg/kg and killed respectively one (**a**) and twenty-four hours (**b**) after the last treatment. Bar graphs represent the mean ± SEM from eight independent determinations for each experimental group. Red dashed line in panel a indicates the y axis limit (100) of the bar chart represented in panel b. Ordinary one-way ANOVA followed by Tukey’s multiple comparisons test, or Kruskal−Wallis one-way ANOVA for ranks followed by Dunn’s multiple comparisons test for non-normally distributed data. * *p* < 0.05, *** *p* < 0.001 vs. 5 mg/kg, ^###^
*p* < 0.001 vs. 15 mg/kg. CBD 5 mg/kg *n* = 8 CBD 15 mg/kg *n* = 8, CBD 30 mg/kg *n* = 8.

**Figure 2 biomedicines-10-01853-f002:**
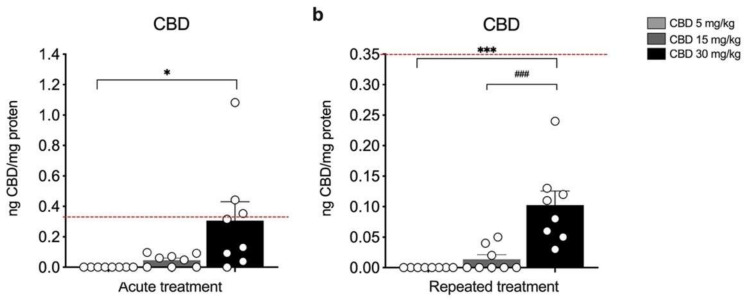
Concentration of CBD in the medial prefrontal cortex (mPFC) of rats treated with a single (**a**) or repeated (**b**) CBD administration at the dose of 5 mg/kg, 15 mg/kg, or 30 mg/kg and killed respectively one (**a**) and twenty-four hours (**b**) after the last treatment. Bar graphs represent the mean ± SEM from eight independent determinations for each experimental group. Red dashed line in panel a indicates the y axis limit (0.35) of the bar chart represented in panel b. Ordinary one-way ANOVA followed by Tukey’s multiple comparisons test, or Kruskal−Wallis one-way ANOVA for ranks followed by Dunn’s multiple comparisons test for non-normally distributed data. * *p* < 0.05, *** *p* < 0.001 vs. 5 mg/kg, ^###^
*p* < 0.001 vs. 15 mg/kg. CBD 5 mg/kg *n* = 8 CBD 15 mg/kg *n* = 8, CBD 30 mg/kg *n* = 8.

**Figure 3 biomedicines-10-01853-f003:**
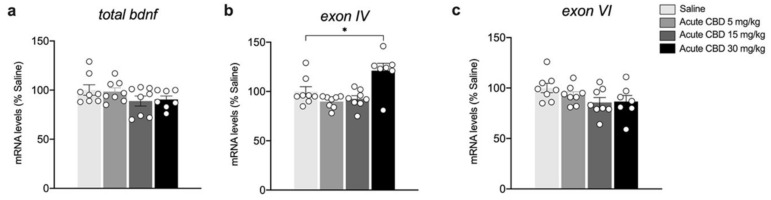
Effects of acute CBD exposure on Bdnf gene expression levels in the mPFC. Rats were treated with a single injection of CBD 5 mg/kg, 15 mg/kg, or 30 mg/kg, and killed one hour after the treatment. *Total Bdnf* (**a**), *Bdnf exon IV* (**b**), and *Bdnf exon VI* (**c**) mRNA levels in mPFC are expressed as percentages of saline-treated rats. Bar graphs represent the mean ± SEM from eight independent determinations for each experimental group. Ordinary one-way ANOVA followed by Tukey’s multiple comparisons test, or Kruskal−Wallis one-way ANOVA for ranks followed by Dunn’s multiple comparisons test for non-normally distributed data. * *p* < 0.01 vs. saline-treated rats. Saline *n* = 8 CBD 5 mg/kg *n* = 8 CBD 15 mg/kg *n* = 8, CBD 30 mg/kg *n* = *7*.

**Figure 4 biomedicines-10-01853-f004:**
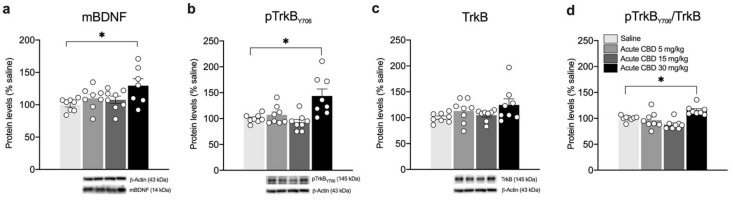
Effects of acute CBD exposure on mBDNF and TrkB receptor protein levels in the mPFC. Rats were treated with a single injection of CBD 5 mg/kg, 15 mg/kg, or 30 mg/kg and killed one hour after treatment. Protein levels of BDNF (**a**), phospho(p)-TrkBY706 (**b**), TrkB (**c**), and of the ratio pTrkB/TrkB (**d**) measured in the homogenate of mPFC are expressed as percentages of saline-treated rats. Below the graphs representative immunoblots are shown for mBDNF (14 kDa), pTrkB Y706 (145 kDa), TrkB (145 kDa), and β-Actin (43 kDa) proteins. Bar graphs represent the mean ± SEM from eight independent determinations for each experimental group. Ordinary one-way ANOVA followed by Tukey’s multiple comparisons test, or Kruskal-Wallis one-way ANOVA for ranks followed by Dunn’s multiple comparisons test for non-normally distributed data. * *p* < 0.05 vs. saline-treated rats. Saline *n* = 8, CBD 5 mg/kg *n* = 8 CBD 15 mg/kg *n* = 8, CBD 30 mg/kg *n* = 8 (mBDNF *n* = 7).

**Figure 5 biomedicines-10-01853-f005:**
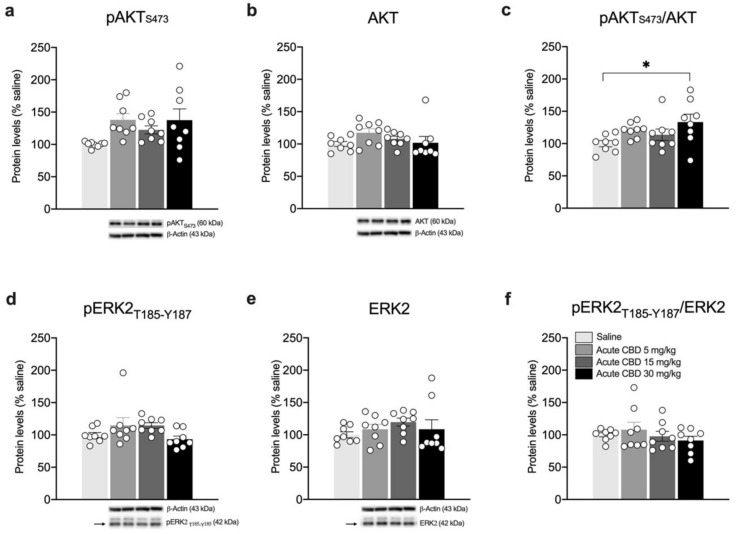
Effects of acute CBD exposure on BDNF-downstream signaling in the mPFC. Rats were treated with a single injection of CBD 5 mg/kg, 15 mg/kg, or 30 mg/kg and killed one hour after treatment. Protein levels of phospho(p)-AktS473 (**a**), Akt (**b**), pAkt/Akt (**c**), pERK2T185-Y187 (**d**), ERK2 (**e**), and pERK2/ERK2 (**f**) measured in the homogenate of mPFC are expressed as percentages of saline-treated rats. Below the graphs, representative immunoblots are shown for pAktS473 (60 kDa), Akt (60 kDa), pERK2T185-Y187 (42 kDa), ERK2 (42 kDa), and β-Actin (43 kDa) proteins. Bar graphs represent the mean ± SEM from eight independent determinations for each experimental group. Ordinary one-way ANOVA followed by Tukey’s multiple comparisons test, or Kruskal-Wallis one-way ANOVA for ranks followed by Dunn’s multiple comparisons test for non-normally distributed data. * *p* < 0.05 vs. saline-treated rats. Saline *n* = 8, CBD 5 mg/kg *n* = 8 CBD 15 mg/kg *n* = 8, CBD 30 mg/kg *n* = 8.

**Figure 6 biomedicines-10-01853-f006:**
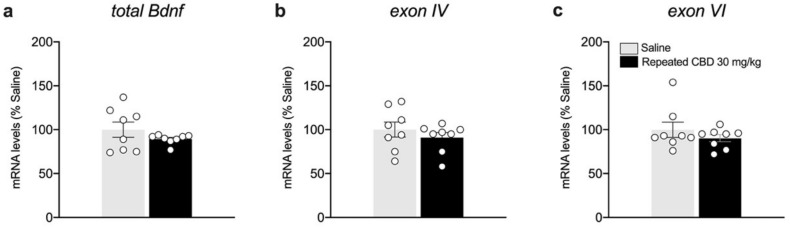
Effects of repeated CBD exposure on Bdnf gene expression levels in the mPFC. Rats were treated with repeated injections of CBD 30 mg/kg for seven days and killed twenty-four hours after the last treatment. *Total Bdnf* (**a**), *exon IV* (**b**), and *exon VI* (**c**) mRNA levels in mPFC are expressed as percentages of saline-treated rats. Bar graphs represent the mean ± SEM from eight independent determinations for each experimental group. Unpaired Student’s *t*-test. Saline *n* = 8, CBD 30 mg/kg *n* = 8.

**Figure 7 biomedicines-10-01853-f007:**
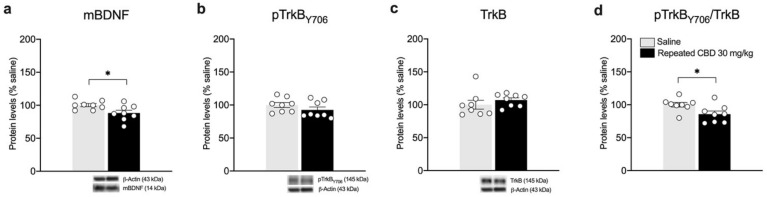
Effects of repeated CBD exposure on mBDNF and TrkB receptor protein levels in the mPFC. Rats were treated with repeated injections of 30 mg/kg for seven days and killed twenty-four hours after the last treatment. Protein levels of BDNF (**a**), phospho(p)-TrkBY706 (**b**), TrkB (**c**), and of the ratio pTrkB/TrkB (**d**) measured in the homogenate of mPFC are expressed as percentages of saline-treated rats. Below the graphs, representative immunoblots are shown for mBDNF (14 kDa), pTrkBY706 (145 kDa), TrkB (145 kDa), and β-Actin (43 kDa) proteins. Bar graphs represent the mean ± SEM from eight independent determinations for each experimental group. Unpaired Student’s *t*-test or Wilcoxon−Mann−Whitney test. * *p* < 0.05 vs. saline-treated rats. Saline *n* = 8, CBD 30 mg/kg *n* = 8.

**Figure 8 biomedicines-10-01853-f008:**
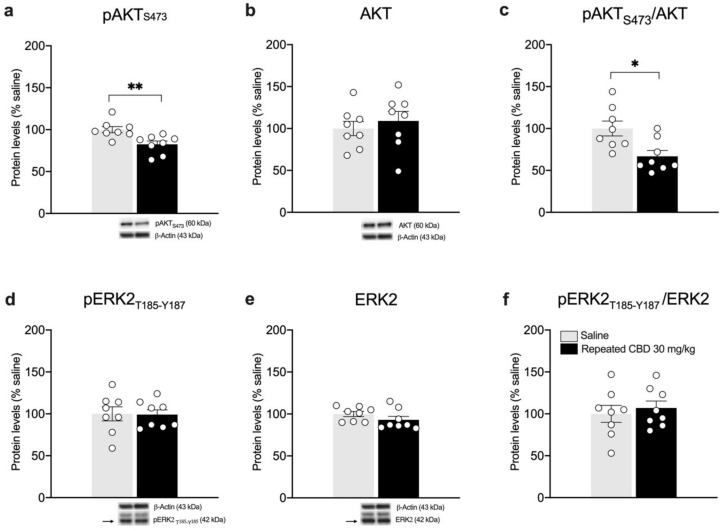
Effects of repeated CBD exposure on BDNF-downstream signaling in the mPFC. Rats were treated with repeated injections of 30 mg/kg for seven days and killed twenty-four hours after the last treatment. Protein levels of phospho(p)-AktS473 (**a**), Akt (**b**), pAkt/Akt (**c**), pERK2T185-Y187 (**d**), ERK2 (**e**), and pERK2/ERK2 (**f**) measured in the homogenate of mPFC are expressed as percentages of saline-treated rats. Below the graphs, representative immunoblots are shown for pAktS473 (60 kDa), Akt (60 kDa), pERK2T185-Y187 (42 kDa), ERK2 (42 kDa), and β-Actin (43 kDa) proteins. Bar graphs represent the mean ± SEM from eight independent determinations for each experimental group. Unpaired Student’s *t*-test or Wilcoxon−Mann−Whitney test. * *p* < 0.05, ** *p* < 0.01 vs. saline-treated rats. Saline *n* = 8, CBD 30 mg/kg *n* = 8.

**Figure 9 biomedicines-10-01853-f009:**
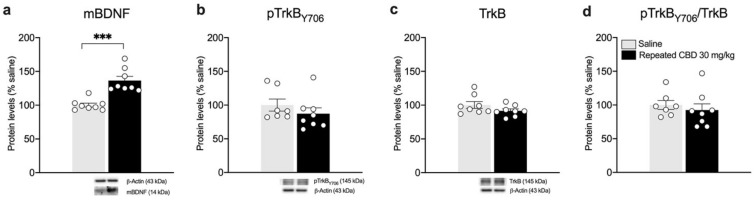
Effects of repeated CBD exposure on mBDNF and TrkB receptor protein levels in the striatum. Rats were treated with repeated injections of 30 mg/kg for seven days and killed twenty-four hours after the last treatment. Protein levels of mBDNF (**a**), phospho(p)-TrkBY706 (**b**), TrkB (**c**), and of the ratio pTrkB/TrkB (**d**) measured in the homogenate of the striatum are expressed as percentages of saline-treated rats. Below the graphs, representative immunoblots are shown for mBDNF (14 kDa), pTrkBY706 (145 kDa), TrkB (145 kDa), and β-Actin (43 kDa) proteins. Bar graphs represent the mean ± SEM from eight independent determinations for each experimental group. Unpaired Student’s *t*-test. *** *p* < 0.001 vs. saline-treated rats. Saline *n* = 8 (pTrkB *n* = 7), CBD 30 mg/kg *n* = 8.

**Figure 10 biomedicines-10-01853-f010:**
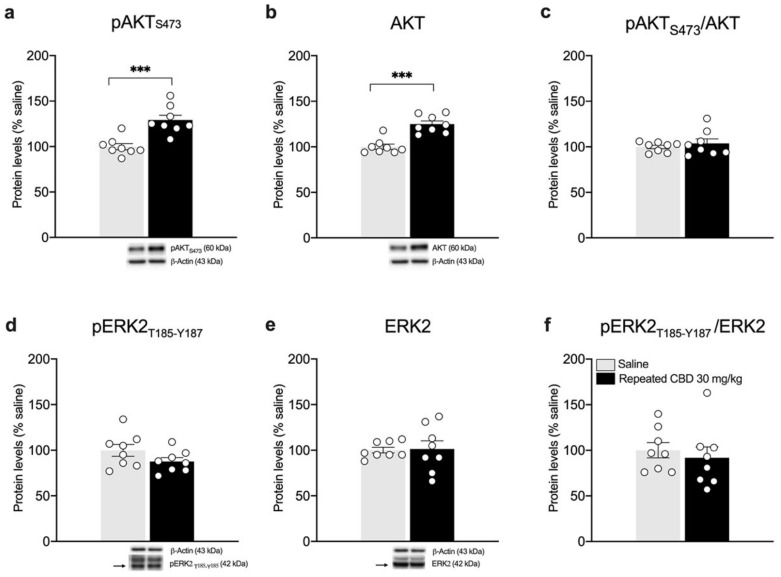
Effects of repeated CBD exposure on BDNF-downstream signaling in the striatum. Rats were treated with repeated injections of 30 mg/kg for seven days and killed twenty-four hours after the last treatment. Protein levels of phospho(p)-AktS473 (**a**), Akt (**b**), pAkt/Akt (**c**), pERK2T185-Y187 (**d**), ERK2 (**e**), and pERK2/ERK2 (**f**) measured in the homogenate of the striatum are expressed as percentages of saline-treated rats. Below the graphs, representative immunoblots are shown for pAktS473 (60 kDa), Akt (60 kDa), pERK2T185-Y187 (42 kDa), ERK2 (42 kDa), and β-Actin (43 kDa) proteins. Bar graphs represent the mean ± SEM from eight independent determinations for each experimental group. Unpaired Student’s *t*-test. *** *p* < 0.001 vs. saline-treated rats. Saline *n* = 8, CBD 30 mg/kg *n* = 8.

**Table 1 biomedicines-10-01853-t001:** Chromatographic gradient used for analysis by LC-MS/MS. Phase A: 0.1% formic acid in water; phase B: methanol.

Time (min)	Flow (mL/min)	% A	% B
0	0.8	40	60
1	0.8	40	60
10	0.8	0	100
12	0.8	0	100
12.10	0.8	40	60
17	0.8	40	60

## Data Availability

Not applicable.

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
