# Peer review of "Single and Repeated Exposure to Cannabidiol Differently Modulate BDNF Expression and Signaling in the Cortico-Striatal Brain Network"

_biomedicines, 2022, doi:10.3390/biomedicines10081853_

Round 1

Reviewer 1 Report

This is an interesting descriptive report of the effects of several doses of CBD in rats on BDNF mRNA in the mPFC and striatum.  The paper is clear and the data are well described and presented. 

Suggestions and questions

1.     The authors should include in the methods the lower limit of detection of their CBD assays in tissue and in plasma.  Without that, it is difficult to interpret the lack of signal in the samples from rats treated with 5 and 15 mg/kg.

2.     It seems that CBD is at higher concentrations in plasma than in the brain.  Can the authors estimate the fraction of CBD that has distributed to the brain at the 30 mg/kg dose?

3.     There are several aspects of the discussion that are difficult to understand or seem off topic. 

a.     Starting on line 6 (Of note…) the authors compare the CBD in brain after a single and multiple injections, then state that such a comparison should not be made since the time of harvest was different in the two studies.  I suggest that the comparison should not be included at all.

b.     On lines 8-9 on page 12, the authors claim their peak concentration of CBD in brain occurred at 1 hour, however, they did not do a time course so cannot know this.

c.     The discussion of fluoxetine in the middle of page 12 is confusing and does not add much, at least as written.  Consider removing or rewriting this for clarity.

4.     The pharmacokinetics of CBD, particularly the fact that it can induce its own metabolism, should be mentioned in the discussion.  Since CBD metabolites were not measured in this study, little can be concluded, but the possibility that multiple exposures to CBD may enhance its metabolism and, therefore, reduce the amount available to enter the brain should be considered.

Author Response

Reviewer #1

General comment:

This is an interesting descriptive report of the effects of several doses of CBD in rats on BDNF mRNA in the mPFC and striatum.  The paper is clear and the data are well described and presented. 

Answer: We thank the reviewer for his/her very positive comments on our manuscript.

Suggestions and questions:

1) The authors should include in the methods the lower limit of detection of their CBD assays in tissue and in plasma.  Without that, it is difficult to interpret the lack of signal in the samples from rats treated with 5 and 15 mg/kg.

Answer: LOD for CBD analysis was calculated and a sentence was added in the materials and methods section (LC-MS analysis).

 2) It seems that CBD is at higher concentrations in plasma than in the brain.  Can the authors estimate the fraction of CBD that has distributed to the brain at the 30 mg/kg dose?

Answer: This is indeed an interesting question. However, we cannot estimate the CBD distributed to the whole brain because we have performed our experiments in different brain areas and not in the whole homogenate. Accordingly, we cannot answer to this interesting question.

3)  There are several aspects of the discussion that are difficult to understand or seem off topic. 

  1. Starting on line 6 (Of note…) the authors compare the CBD in brain after a single and multiple injections, then state that such a comparison should not be made since the time of harvest was different in the two studies.  I suggest that the comparison should not be included at all.
  2. On lines 8-9 on page 12, the authors claim their peak concentration of CBD in brain occurred at 1 hour, however, they did not do a time course so cannot know this.
  3. The discussion of fluoxetine in the middle of page 12 is confusing and does not add much, at least as written.  Consider removing or rewriting this for clarity.

Answer: We thank the reviewer for these comments. Specifically,a)     We agree with the reviewer comment and have removed the unproper comparison (page 11).b)     We agree with the reviewer’s note and have removed the comparison accordingly (page 12).c)     We agree with the reviewer and have removed the part including fluoxetine. 

4) The pharmacokinetics of CBD, particularly the fact that it can induce its own metabolism, should be mentioned in the discussion.  Since CBD metabolites were not measured in this study, little can be concluded, but the possibility that multiple exposures to CBD may enhance its metabolism and, therefore, reduce the amount available to enter the brain should be considered. 

Answer: We thank the reviewer for the suggestion. The following comment regarding the interaction of CBD with CYP450 and the formation of CBD-derived metabolites was included in the discussion.

“As already revised by Lucas et al. (2018) and Ujvàry et al. (2016), CBD is mainly metabolized by iso-enzymes CYP2C19 and CYP3A4 at hepatic level. Several studies involving rodent models showed that CBD is mainly excreted in the intact or glucuronide form, while the major metabolites are hydroxylated derivatives and their glucuronide conjugates. For this reason, hepatic passage could be involved in the reduction of CBD plasmatic concentration during repeated exposure. However, the pharmacology of hydroxylated metabolites is still poorly investigated.”

Here the references added in the tet:

  1. Lucas, C.J.; Galettis, P.; Schneider, J. The pharmacokinetics and the pharmacodynamics of cannabinoids. Br J Clin Pharmacol 2018, 84, 2477-2482, doi:10.1111/bcp.13710.
  2. Ujvary, I.; Hanus, L. Human Metabolites of Cannabidiol: A Review on Their Formation, Biological Activity, and Relevance in Therapy. Cannabis Cannabinoid Res 2016, 1, 90-101, doi:10.1089/can.2015.0012.

Reviewer 2 Report

The aims and hypotheses of the study and findings were presented in a logical and coherent manner, and they allowed for convincing arguments to be developed. The introduction was well-written with appropriate fundamental background on the driven hypothesis for the purpose of the study. The presentation of results was appropriate, and they were at high standard levels.

 Minor comments:

1.     A control group with saline injection should be added for results presented in figure 1.

2.     The statistical tests used assume a normal distribution. Please state test of normality used. If the data are not normal, please use a non-parametric alternative.

3.     If n is too small to determine normality (n<6) please use a non-parametric alternative. These include better reference to ARRIVE guidelines; justification for statistical analysis of a (small number) of dataset n<6. There are also other studies which could be referenced here, for example doi: 10.1016/j.neuropharm.2018.08.037.

4.     Relative expression needs to be clearly defined. What is it relative to?

5.     Please ensure that the number of biological and/or technical replicates ("n numbers") are consistently reported in all figure legends.

6.     All methods should be of sufficient detail to allow replication, even for those procedures that may be considered standard. Referring to previously published procedures or manufacturer protocols is accepted; however, any deviations should be detailed in the text. Please confirm that all methods adhere to this standard.

Author Response

Reviewer #2

General comment:

The aims and hypotheses of the study and findings were presented in a logical and coherent manner, and they allowed for convincing arguments to be developed. The introduction was well-written with appropriate fundamental background on the driven hypothesis for the purpose of the study. The presentation of results was appropriate, and they were at high standard levels.

Answer: We thank the reviewer for his/her positive comments on our manuscript.

 Minor comments:

1) A control group with saline injection should be added for results presented in figure 1. 

Answer: We thank the reviewer for this note. However, we cannot have a saline group for results presented in figure 1 as this group will have undetectable levels of CBD.

2) The statistical tests used assume a normal distribution. Please state test of normality used. If the data are not normal, please use a non-parametric alternative. 

Answer: We thank Reviewer 2 for this consideration. We have amended the request by calculating per each set of data the normality of residuals by means of the Shapiro-Wilk and the Kolmogorov–Smirnov tests. Accordingly, data with a non-normal distribution were re-analyzed with the Kruskall-Wallis one-way ANOVA for ranks followed by Dunn’s multiple comparisons test (which replaces the ordinary one-way ANOVA for normal distributed data sets), or with the Wilcoxon-Mann-Whitney test (which replaces the unpaired student’s t test for normal distributed data sets), respectively. All the statistical specifics relative to the analysis of normality are presented in the supplementary tables 1-2-3, and we have added the proper description either in the Materials and methods section and in the figure legends.

3) If n is too small to determine normality (n<6) please use a non-parametric alternative. These include better reference to ARRIVE guidelines; justification for statistical analysis of a (small number) of dataset n<6. There are also other studies which could be referenced here, for example doi: 10.1016/j.neuropharm.2018.08.037.

Answer: We thank Reviewer 2 for this consideration however all the presented sets of data are characterized by n=8. Even when outliers are excluded from the mean the groups are characterized at least by n7and these exceptions are now properly explained in the figure legends. Accordingly, we did not have the statistical necessity to cite the suggested reference.

4) Relative expression needs to be clearly defined. What is it relative to?

Answer: We searched throughout the manuscript (both text and supplementary material) but we have not fountd the sentence including ’relative expression’.

5) Please ensure that the number of biological and/or technical replicates ("n numbers") are consistently reported in all figure legends.

 Answer: We thank the reviewer for this note and have amended the manuscript accordingly.

6) All methods should be of sufficient detail to allow replication, even for those procedures that may be considered standard. Referring to previously published procedures or manufacturer protocols is accepted; however, any deviations should be detailed in the text. Please confirm that all methods adhere to this standard.

Answer: We thank Reviewer 2 for having raised this point. As we stated in the methods, we confirm that all the procedures described were performed according to the cited references, and every modification relative to the reference are explicated in the text.